

# Evaluating the Atibaia River Hydrology using JULES6.1

Hsi-Kai Chou[1], Ana Maria Heuminski De Avila [2], Michaela Bray[1]

[1]School of Engineering, Cardiff University, Cardiff, CF24 3AA, UK

[2]Center for Meteorological and Climate Research Applied to Agriculture (CEPAGRI) at the State University of Campinas, Campinas, SP 13083-871, Brazil

*Correspondence to*: Hsi-Kai Chou (ChouH2@cardiff.ac.uk)

**Abstract.**

Land surface models such as the Joint UK Land Environment Simulator (JULES) are increasingly used for hydrological assessments because of their state-of-the-art representation of physical processes and versatility. Unlike statistical models and AI models, the JULES model simulates the physical water flux under given meteorological conditions, allowing us to understand and investigate the cause and effect of environmental processes changes. Here we explore the possibility of this approach using a case study in the Atibaia river basin, which serves as a major water supply for metropolitan regions of Campinas and São Paulo, Brazil. The watershed is suffering increasing hydrological risks, which could be attributed to environmental changes, such as urbanization and agricultural activity. The increasing risks highlight the importance to evaluate the land surface processes of the watershed systematically. We explore the use of local precipitation collection complement with multiple sources of global reanalysis data to simulate the basin hydrology. Our results show that the coarse resolution of rainfall data is the main reason to reduce model performance. Despite this shortcoming, key hydrological fluxes in the basin can be represented by the JULES model simulations.

## 1 Introduction

The Atibaia river basin serves as a major water supply for Campinas and São Paulo (Demanboro, Laurentis & Bettine, 2013; Nobre et al., 2016). The basin is subjected to human impacts such as urbanization and agricultural activities. Increasing hydrological risks have emerged with historic floods in 2009 and 2010 (SANTEIRO & CAMPOS, CELSO DAL RÉ CARNEIRO, 2015), and drought in 2014 and 2015 (Marengo et al., 2015; Nobre et al., 2016), which highlight the importance to evaluate the hydrology systematically.

Models are simplified and imperfect representations of a real-world system, which can help predict system behaviour and understand various processes. One of their main applications is to make predictions about the potential impact of changes. Hydrological models are an essential tool in hydrological science and catchment management for evaluating the hydrological impacts of climate change or land-use and land-cover change (Buytaert & Beven, 2011).

A few research activities have investigated the Atibaia river basin due to its importance in the water supply. At present, the Paraná Meteorological System (Simepar) operates a predictive model system using three types of models in the Atibaia





river basin to generate flow forecasts in the upcoming seven days. The models included a conceptual model, known as "Sacramento", adopted by the National Weather Service (NWS); a multilinear regressive model based on statistical theory and a deep learning model based on artificial intelligence (Prochmann, 2019). On the other hand, applying the Bayesian Error Modelling methodology through Monte Carlo Simulation via Markov Chains (MCMC) generates a probabilistic

forecast, indicating high and low chances of future flow occurring in a given range of values. Through simulations of future behaviours, this system measures, in percentages, the probability that the flow will be above or below critical levels in time and space, which is then used for decision-making in the management of water resources. However, the Bayesian Error Modelling methodology model is not in the public domain. Therefore, there is still a need to develop a hydrological model to support regional studies.

Physically-based hydrological models are often used to simulate the physical water flux under given meteorological conditions, allowing us to understand and investigate the cause and effect of environmental processes changes. A commonly used Soil Water Assessment Tool (SWAT) has been operated for flow and sediment estimation for the Atibaia river basin (dos Santos, de Oliveira & Mauad, 2020). The accuracy of the model highly depends on the model structure, availability and quality of input data. Also, local calibration is usually required due to the few empirical approximations in each model.

The JULES model was developed from the Met Office Surface Exchange Scheme (MOSES) by the UK Met Office (Cox et al., 1999). It can be coupled to an atmospheric global circulation model but is also used as a standalone land surface model which simulates the fluxes of carbon (Clark, D. B. et al., 2011), water, energy and momentum (Best et al., 2011) between the land surface and the atmosphere. The model is driven by a large dataset of hydrometeorological variables using a physically-based approach, which has been increasingly used for hydrological assessment (e.g. Le Vine et al., 2016;

Zulkafli et al., 2013). Therefore, we examine JULES's model ability to simulate the land surface processes of the Atibaia watershed.

## 2 Methods and data

### 2.1 The JULES model

JULES simulates the energy exchange between different land surface processes detailly described by Best et al. (2011)

and Clark et al. (2011). For each study site, we classified the land cover into five vegetated Plant Functional Types (Harper et al., 2018): including tropical broadleaf evergreen trees (BET-Tr), needle-leaf evergreen trees (NET), C3 grasses (C3), C4 grasses (C4), evergreen shrubs (ESH), and a non-vegetated: bare soil (BS) using MODIS data (Friedl & Sulla-Menashe, 2015). Distinct parameters are used to calculate the energy balance of surface temperatures, short-wave and long-wave radiative fluxes, sensible and latent heat fluxes, ground heat fluxes, canopy moisture contents, snow masses and snow

melting rates for each surface type in a grid-box.

The sub-grid surface heterogeneity is presented using a tiled model upon a shared 4-layer soil column with a thickness of 0.1, 0.25, 0.65, and 2.0 m from top to bottom. In JULES, precipitation is intercepted by the canopy storage, then





partitioned into surface flow and infiltration into the soil based on the Hortonian infiltration excess mechanism. In our model setup, we have calculated saturation excess flow by first using the Probability Distributed Model (PDM) described by Moore (1985), with the sub-grid distribution of soil moisture described by a probability function (Clark, Douglas B. & Gedney, 2008); and then again using the TOPMODEL approach (Beven and Kirkby, 1979). An instantaneous redistribution of soil moisture is assumed for the infiltration following the Darcy–Richards diffusion equation. The gravity drainage generates the subsurface flow at the lower boundaries. Soil hydraulic characteristics can be estimated using the relationship of Brooks & Corey (1964) or a more robust formulation of Van Genuchten (1980). The required soil parameters are obtained by using pedotransfer functions (PTFs) of Hodnett & Tomasella (2002), which generates parameters from physics and chemical properties of soil obtained from a large-scale soil database (FAO/IIASA/ISRIC/ISSCAS/JRC, 2012). We evaluated the sensitivity of hydrological parameters using 1) the PDM and 2) TOPMODEL (Table 1) to determine the most suitable approach to describe the Atibaia basin hydrology.

For each sub-basin (17 sub-basins covering 128.8 km$^2$ in average, Figure 1), the surface ($Q_{surface}$) and sub-surface ($Q_{subsurface}$) runoff fluxes are simulated with rainfall data from the nearest monitoring station of Campinas-IAC (Campinas, Atibaia, and Nazare Paulista). The year with high missing rainfall records (e.g. 2012 and 2015) is replaced by the nearest DAEE station's time series. The runoff fluxes simulated by the JULES model require an external river routing model for a reasonable comparison to observed river flows (Best et al., 2011). In this study, we applied a simple delayed function to account for the routing delay in the river discharge ($Q_{sim}$) in each timestep (t). For each basin, the delay time ($t_i$) is dividing the distance to the outlet by flow speed (C). We set the flow speed constantly as the average flow speed of 0.36 m/s (from 2009 to 2019).

$$Q_{sim,t} = \sum_{i=1}^{n} \left( Q_{surface,t-t_{i1}} + Q_{subsurface,t-t_{i2}} \right); t_{i1} = \frac{d_i}{C_{surface}}; t_{i2} = \frac{d_i}{C_{subsurface}}$$

We evaluated the sensitivity of hydrological parameters of PDM and TOPMODEL to determine the most suitable model for the Atibaia river basin using the simulated results from the first year. Soil depth (dz_pdm), shape factor for the pdf (b_pdm), the fraction of maximum storage (s_pdm) is evaluated for PDM, and the maximum allowed depth to the water table (zw_max), the maximum possible value of the topographic index (ti_max) for the TOPMODEL.

### 2.2 Study Region and data

This study explores the hydrology of the Atibaia River Basin. The altitude of the catchment ranges between 530 m and 1818 m; it is located between the coordinates 22°40′ and 23°20′ S and 47°20′ and 46°00′ W in south-eastern Brazil, covering an area of 2816.4 km$^2$ (Figure 1). The modelling results cover 2075.2 km$^2$ effectively since Cachoeira Dam and Atibainha Dam intercept the upstream flow, and the monitor station does not cover part of the lower basin. The primary soil types in this area are Ferralsols, Acrisols, Leptosols, and Cambisols (FAO/IIASA/ISRIC/ISSCAS/JRC, 2012; Ottoni et al., 2018;





Rossi, 2017). The primary land cover is rural (53.0%), followed by forest (27.6%), and then urban (12.0%). Higher
percentages of forest are found in the upper basin, whereas urban areas concentrate in the lower basin.

The study region's rainfall presents a seasonal pattern with rainy summer and dry winter (Cavalcanti et al., 2017; Dias
et al., 2013). The rainfall regimes are influenced by the passage and frontal systems' intensity (Maddox, 1983; Silveira et al.,
2016). The maximum precipitation occurs during the austral summer associated with the South Atlantic Convergence Zone
(SACZ) and in the winter predominates the high pressing system of the South Atlantic (Jones & Carvalho, 2013). Time
series of rainfall from 2009 to 2019 are provided by Campinas Agronomic Institute (Campinas-IAC) and the Department of
Water and Electricity (DAEE) situated in São Paulo. The temperature, specific humidity, and surface pressure are observed
by the Center for Meteorological and Climate Research Applied to Agriculture (CEPAGRI). The air temperature is elevation
adjusted with the lapse rate ($\gamma$) 1.4 °C per 100 meters (Figure 2) obtained from Campinas-IAC and CEPAGRI data during
the study period.

        Other meteorological data required include downward short-wave radiation, long-wave radiation, and wind speed, all
of which are extracted from the NCEP-DOE Reanalysis II dataset (Kanamitsu et al., 2002). The dataset is available on a T62
Gaussian grid with 192 x 94 points (approximately 2° scale) and provides a 6-hourly temporal resolution from 1979/01 up to
the present. The 6-hourly resolution was disaggregated into hourly data using linear interpolation.

        The abstract and release data of the dams are obtained from the Basic Sanitation Company of the State of Sao Paulo
(SABESP, 2020). River flow observations from the basin's outlet (station 4D-009) measured by DAEE are used for model
calibration and validation.

## 2.3 Model evaluation

        We evaluated the modelling daily flow using the river flow observations measured by DAEE by summarizing the
simulated time series into hydrological indices, including, Baseflow index (BFI), daily flow variation (Qvar), and Nash–
Sutcliffe model efficiency (NSE), per cent bias ($P_{BIAS}$), and coefficient of determination ($r^2$).

        BFI defines the ratio of baseflow ($Q_{base}$) to the total flow. For dry weather runoff assessment, we separated the baseflow
from the total flow using the two-parameter algorithm from Chapman (1999) with a filter parameter of 0.085 (Ochoa-
Tocachi, Buytaert & De Bièvre, 2016).

$$\text{BFI} = \frac{Q_{base}}{Q} \tag{1}$$

We assess the flow stability using Qvar, which is the standard deviation ($\sigma_Q$) divided by its mean value ($Q_{mean}$).

$$\text{Qvar}: \frac{\sigma_Q}{Q_{mean}} \tag{2}$$



We evaluated the overall model performance using the Nash Sutcliffe Efficiency (NSE), percent bias (PBIAS), and coefficient of determination ($r^2$):


$$\text{NSE} = 1 - \frac{\sum_t^N (Q_{mod,t} - Q_{obs,t})^2}{\sum_t^N (Q_{obs,t} - Q_{obs,mean})^2} \tag{3}$$

$$P_{BAIS} = \left( \frac{\sum_t^N Q_{obs,t} - \sum_t^N Q_{mod,t}}{\sum_t^N Q_{obs,t}} \right) \tag{4}$$

$$r^2 = \frac{\left[ \sum_t^N (Q_{obs,t} - Q_{obs,mean})(Q_{mod,t} - Q_{mod,mean}) \right]^2}{\sum_t^N (Q_{obs,t} - Q_{obs,mean})^2 \sum_t^N (Q_{mod,t} - Q_{mod,mean})^2} \tag{5}$$

## 3 Results and Discussion

### 3.1 Sensitivity analysis

We evaluated the sensitivity of hydrological parameters using 1) the PDM and 2) TOPMODEL. For PDM, we found lower flow simulated with increasing soil depth (dz_pdm). However, the average flow only reduced by 2.5 percent when soil depth was increased from 0.8 to 2.0. In contrast, we found that the shape factor (b_pdm) and minimum soil water content (s_pdm) have a higher impact on the simulated flow (Figure 3a-c). When we increased the minimum soil water content, the simulated flow is reduced with more water to be held on the soil, but it gradually alters the hydrograph. The average flow is

reduced by 17.5 percent when the minimum soil water content increased from 0 to 4. We found gradual changes in the flow regime when adjusting the shape factor and minimum soil water content, whereas a higher value of the shape factor can change the flow into a subsurface flow-dominated regime. We found that the shape factor of 0.5 best describes the flow in the study basin, as the higher shape factor simulates a more gradual flow regime (Clark, Douglas B. & Gedney, 2008), which lower the peak level flow estimation and hence increased baseflow generations.

For TOPMODEL, we found a merely change when modifying the maximum allowed depth to the water table (zw_max) within the 5 and 8-meter range. However, the maximum possible value of the topographic index (ti_max) can affect the simulation. There is no change in the results until the value been set below 4. Afterwards, the lower parameter value has reduced the average simulated flow (Figure 3d). Peak flow was reduced when lower ti_max was set. We found that the default TOPMODEL better simulated the average flow than the adjusted parameter values, and it also marked the highest

NSE score. Therefore, we run the full-time series of modelling with the default TOPMODEL setup.

### 3.2 Hydrological modelling using the JULES model

Table 2 summarize the evaluation of the JULES model in the Atibaia river basin (4D-009) for the calibration (2009-2013) and the validation period (2014-2019). The calibration period presents more intense rainfall, which led to higher flow than the validation period. The TOPMODEL shows better modelling performance (NSE: 0.715, R$^2$: 0.708) in the calibration





period than the PDM (NSE: 0.622, $R^2$: 0.707). The average modelling flow is close to the observed values (TOPMODEL: +4.15%; PDM: +0.36%) for the calibration period, whereas the higher flow was estimated for the validation period (TOPMODEL: +24.38%; PDM: +27.45%). We attributed one possibility to the lower rainfall during the validation period (1271 mm/year) than during the calibration period (1351 mm/year). The lower intense rainfall has reduced the average flow by 35 per cent, which is more than the expected model simulation (23/18 per cent reduction using TOPMODEL/PDM). The

model might lower estimate the evaporation under the condition of the prolonged dry. Nevertheless, the performance (NSE TOPMODEL: 0.669; PDM: 0.457) is still high in the validation period.

Figure 4 shows the modelling performance of daily flow yearly using TOPMODEL. The modelling performance is over 0.5 in 7 out of 11 years. The highest modelling performance is simulated in 2010 (NSE= 0.893), whereas negative scores in 2014 and 2019. The lowest model performance was simulated in 2014, as we found the simulated peak flow is much higher

than the observed values. Uncertainty in rainfall data could be the main driver for the gap. Our research mainly uses rainfall data from 3 stations (complimentary by other 5 DAEE stations) since continuous time-series data is hard to obtain in the study region. Despite the highly variated data, the simulation shows that it is still representative of the significant modelling period. However, coarse data resolution could amplify under/overestimated rainfall in a single site (i.e. underestimated flow in 2012). In 2010, most of the flow is reasonably simulated (Figure 5a), with the recession of flow are modelled faster than

observed at the end of January. In 2012, the model's peak flows were underestimated from May to August (Figure 5b), while the modelling flow is close to the observed values in the rest of the period. In 2014, the hydrograph showed underestimated flow during the whole year (Figure 5c). The magnitude of flow is the main reason for the low modelling performance.

Despite these shortcomings, our results show that it is possible to use the JULES model for hydrological simulation in the Atibaia river basin. The model performance for daily flow is higher than the SWAT model's estimation (dos Santos, de

Oliveira & Mauad, 2020). However, both research pieces have pointed out that rainfall uncertainty is the primary reason for reduced model performance. There is a possibility for the model to be further improved once more adequate rainfall data is available.

## 4 Conclusions

We implemented the JULES6.1 model in the Atibaia river basin for hydrological estimation to evaluate the model

performance. We evaluated the sensitivity of hydrological parameters and calibrated the model to select the most suitable approach for the study region. We find that the default TOPMODEL can reasonably estimate the flow. For PDM, improving modelling performance can be achieved after calibration as these processes are sensitive to the hydrological parameters. Our results show that the JULES setup can detect most peak events and reasonably estimates baseflow. However, the uncertainty of rainfall data could be the primary driver for lower model performance in some period of higher rainfall variation.

Nevertheless, our results suggest that it is possible to use the JULES model for hydrological evaluation in the Atibaia river





basin. A more refined and higher quality of rainfall observation can be the fundamental drive to improve the modelling performance further.

**Code and Data availability**

This work was based on a version of JULES6.1. The instruction to run JULES is available from the JULES FCM repository https://code.metoffice.gov.uk/trac/jules/wiki/WaysToRunJules
The configuration, code, and datasets for this research are available from

https://doi.org/10.5281/zenodo.5147879

**Author contribution**

HKC and MB led the writing and development of the manuscript. AMHdA processed the data and description of the study area. HKC developed the model and performed the simulations. All the authors contributed to the development of ideas and to the reflection process.

**Acknowledgement**

This work was funded by HEFCW GCRF Small Project: SP93 - Pilot flood and drought forecasting and early warning system for Atibaia River Basin.

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



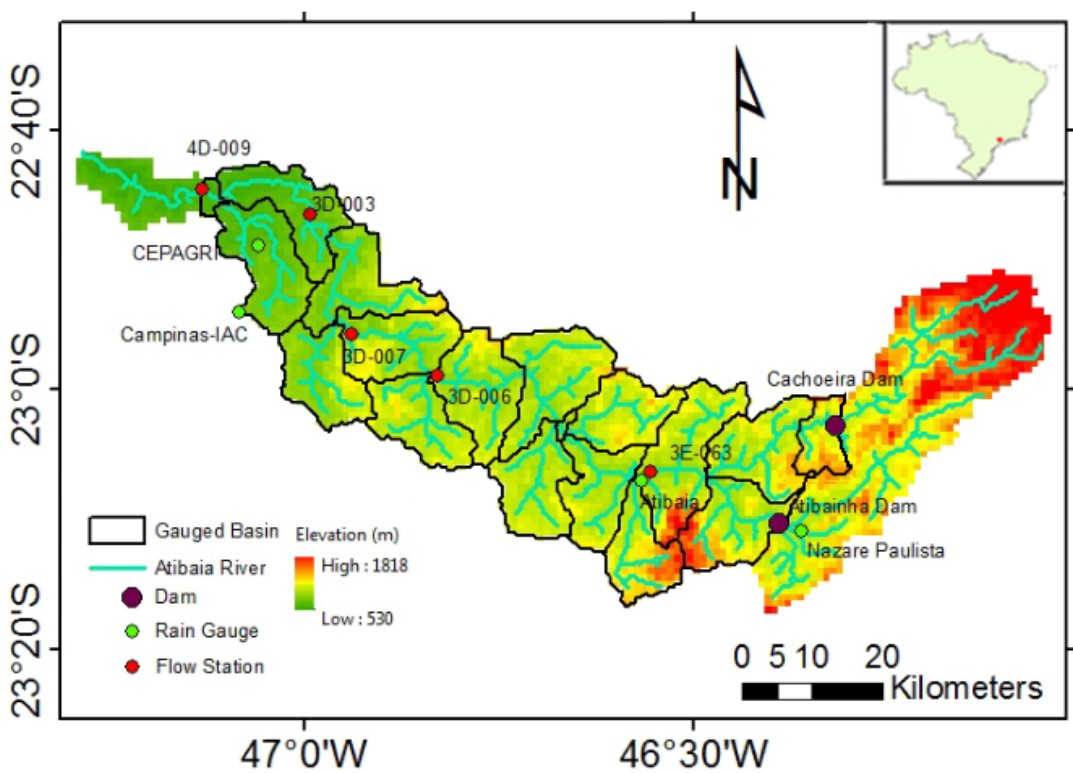


**Figure 1: Atibaia river basin and the location of the rain gauges, flow monitoring stations, and dams.**

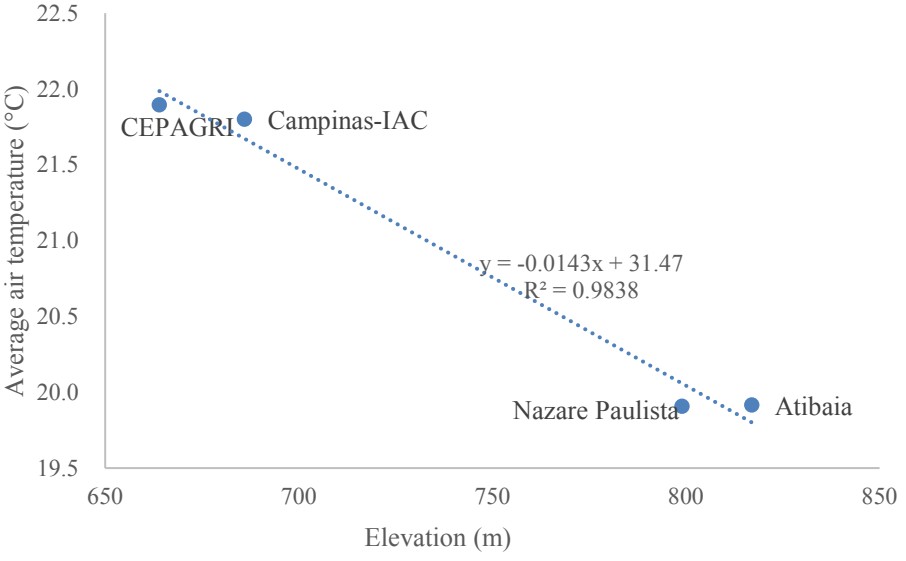

**Figure 2. Average air temperature from 2009-2019 related to the site elevation.**





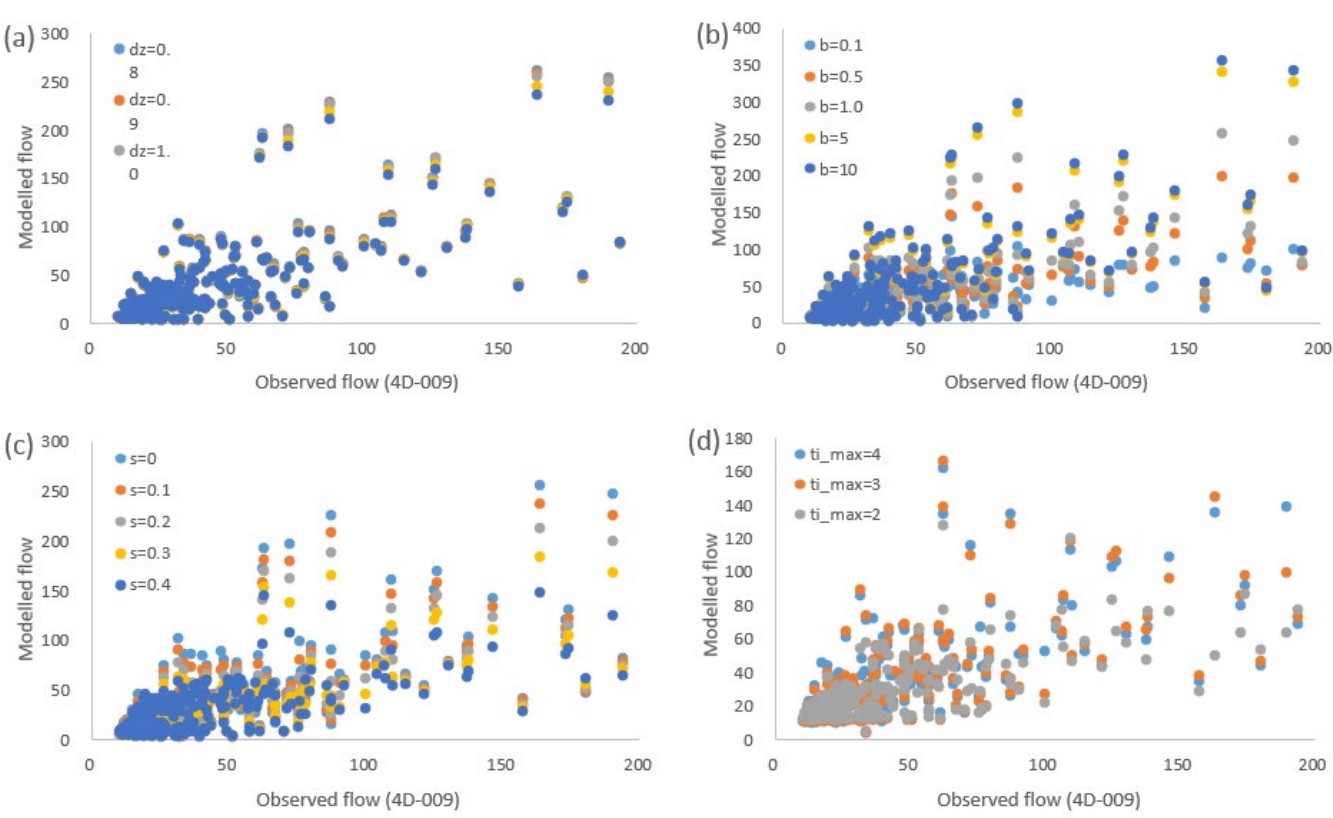

**Figure 3. Sensitivity analysis of a) soil depth b) shape factor c) threshold of minimum soil water content, using PDM. d) topographic index, using TOPMODEL.**

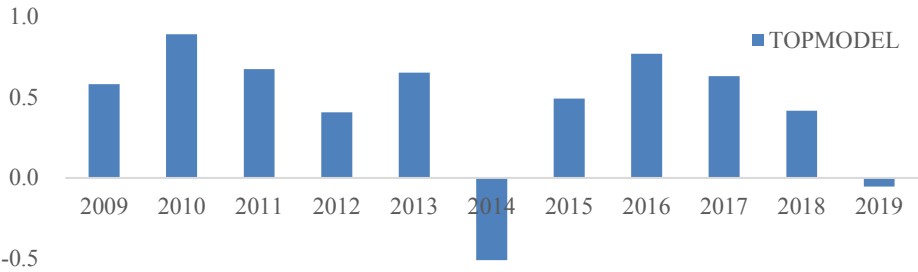

**Figure 4. NSE score of modelling daily flow summarised by year (TOPMODEL).**



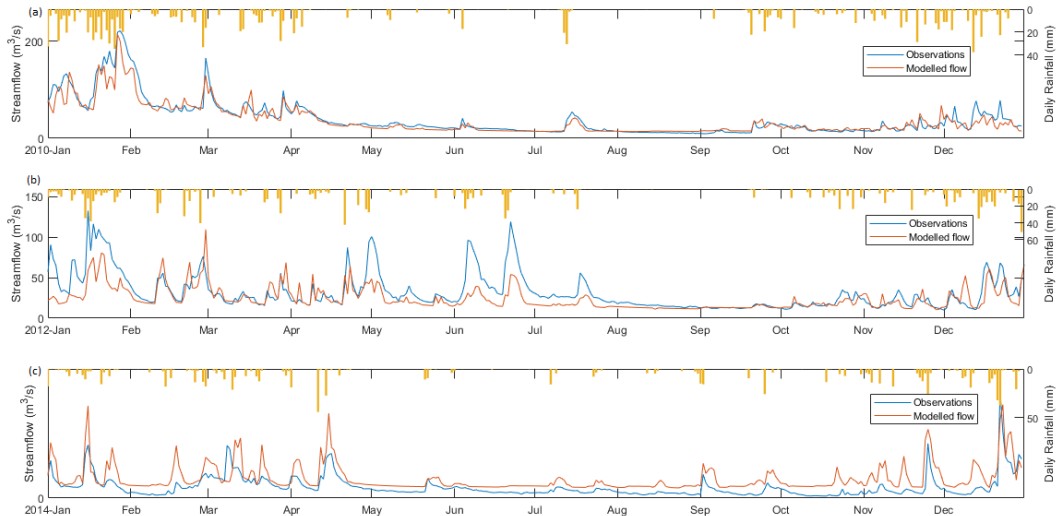

**Figure 5. Daily modelled flow, observed flow, and rainfall in the year (a) 2010, (b) 2012 and (c) 2014.**





Table 1. Sensitivity analysis of hydrological parameters using 1) PDM and 2) TOPMODEL. Default parameter values underlined.

| Parameter | Definition | Sensitivity analysis values |
|---|---|---|
| | PDM | |
| dz_pdm | The depth of soil considered by PDM (m) | 0.8, 0.9, 1.0, 1.1, 1.2 |
| b_pdm | Shape factor for the pdf | 0.1, 0.5, 1, 5, 10 |
| s_pdm | Minimum soil water content below which there is no surface runoff saturation excess production by PDM | 0, 0.1, 0.2, 0.3 |
| | TOPMODEL | |
| zw_max | The maximum allowed depth to the water table (m) | 5, 6, 7, 8 |
| ti_max | The maximum possible value of the topographic index. | 10, 5, 4, 3, 2 |

Table 2. Hydrological summary indices as calculated from the observed (4D009) and modelled flow time series using 1) TOPMODEL and 2) PDM. Calibration period: 2009-2013, validation period: 2014-2019. BFI: baseflow index, Qvar: variance of flow. NSE: Nash Sutcliffe Efficiency. $R^2$: Variance. Pbias: Percentage of bias.

| | Observations | | TOPMODEL | | PDM | |
|---|---|---|---|---|---|---|
| | Calibration | Validation | Calibration | Validation | Calibration | Validation |
| Rainfall [mm/year] | 1351 | 1271 | | | | |
| Average flow [m3/s] | 33.25 | 21.53 | 34.63 | 26.78 | 33.37 | 27.44 |
| BFI | 0.748 | 0.702 | 0.767 | 0.739 | 0.745 | 0.700 |
| Qvar | 0.942 | 1.103 | 0.810 | 0.785 | 1.121 | 1.056 |
| NSE | | | 0.715 | 0.669 | 0.622 | 0.457 |
| $R^2$ | | | 0.708 | 0.703 | 0.707 | 0.658 |
| Pbias | | | -4.15 | -24.38 | -0.36 | -27.45 |