# Peer review of "Evaluating the Atibaia River Hydrology using JULES6.1"

_Geoscientific Model Development, 2021_

## Referee Comment (RC1)

**Review of 'Evaluating the Atibaia River Hydrology using JULES6.1'**

This paper presents results of applying JULES vn 6.1 to the Atibaia river catchment in Brazil. The authors compare modelled river flow predictions with observed values, comparing the two saturation excess schemes available in the JULES code; PDM and TOPMODEL. The authors perform a sensitivity analysis of parameters within PDM and TOPMODEL and conclude that either option can be used to satisfactorily estimate river flow. The authors suggest that better rainfall drivers would improve the model results further. While this is likely true, I feel that consideration of other sources of error would also be appropriate – in particular the relatively coarse resolution of the JULES runs and the choice of river routing model. I also think the manuscript would benefit from some clarifications, particularly in the results and discussion section. Please see below for specific comments.

**Comments:**

Please make sure the citation style is consistent throughout.

Line 11 (and elsewhere) the phrase 'environmental processes changes' is a bit confusing – please could you rephrase this?

Line 16. I disagree with the statement that 'Our results show that the coarse resolution of the rainfall data is the main reason to reduce model performance'

Line 34. I'm not clear what the relevance of the MCMC model is here – is this just another example of alternative modelling methods? Please clarify.

Line 48. I would describe the driving data for JULES as 'meteorological', rather than 'hydrometeorological'.

Line 56. Does the region really have both C3 and C4 grasses? And, later in the paper you mention that 12% of the catchment is urban. Can you comment on that here?

Line 61. Please make sure you are clear in the distinction between precipitation and throughfall.

Line 64. Could you make it a bit clearer that you have compared results from using PDM and TOPMODEL?

Line 71. What is the spatial scale over which you have calculated the soil physics parameters from textures in the HWSD? IS it representative of the subcatchments you have used?

Line 74 (or elsewhere) can you make it clear that you have run JULES with each subcatchment as an effective grid box? The more usual approach is to divide an area of interest into approximately square grid boxes. (I don't think there's any problem with doing it by subcatchment, but it needs to be clear!)

Line 76. Can you make clearer how rainfall data are substituted in the case of missing data? And can you give the reader an idea of how representative the rainfall data are likely to be for each catchment?

Line 80. Can you justify using this simple river routing model rather than RFM or similar, as is more usual? Should the equation in this paragraph be labelled equation (1)? Also please make sure you define all the symbols in all equations in the paper – adding in units can also be really helpful to the

reader. Should there be two different values of C for surface and subsurface flows? How was the value for this chosen?

Line 98. Can you make this clearer?

Line 108. Can you explain how you account for the effect of dams?

Lin 117. Why do you use a value of 0.085 here? I would suggest that equation (1) is not necessary.

Line 120. Which quantity is this the standard deviation of? Please make sure all symbols used are clearly defined – also for the equations from line 125.

Line 125. Please can you make clear to the reader what these metrics mean – i.e. what constitutes a 'good score' etc?

Line 134 (and elsewhere). Can you please make clearer what you mean by 'gradually alters the hydrograph'?

Line 137. You state that the flow is changed to a subsurface flow-dominated regime, but is this shown in the results section?

Line 139. The choice of $b\_pdm = 0.5$ is also not demonstrated in the results shown.

Figure 3. Are these points mean flows? Daily/hourly data points? Please clarify. I would also suggest adding a 1:1 line and/or plotting these with a square aspect ratio to make the relationship clearer. The caption of 3.a is also a little tricky to read.

Line 143. Is this conclusion shown in figure 3d as stated? Please make this clearer if so.

Line 143. Please justify the statement that peak flow was reduced for a lower $ti\_max$.

Lin 150.Please comment on the fact that although TOPMODEL scores better in terms of NSE and $R^2$, the bias is worse.

Line 155: Please reword this as the meaning is unclear

Line 157 and throughout. Please be specific about which metric you are referring to by 'modelling performance' (e.g. NSE here)

Line 160. You state that uncertainty in rainfall data **could** be the main driver for the gap. I agree that it could be, but could you comment on other sources of error and/or uncertainty?

Line 161. What do you mean by 'complimentary' here?

Line 162. Can you rephrase this sentence? Not sure what is meant by 'variated' or 'significant modelling period'.

Line 164 – 167. Can you make sure all of these statements are reflected in the results data shown?

Line 169. You compare to SWAT results here; can you show some results or otherwise justify this assertion?

**Small things**:

Line 15. Suggest changing 'to evaluate' to 'of evaluating'

Line 49. Suggest changing 'which' to 'and'

Line 54. 'detailly' is not a word, please replace

Line 61. Not sure 'presented' is the right word here – could you reword this?

Line 113. Suggest changing 'modelling' to 'modelled'

Line 130. Suggest replacing 'using' with 'in'.

Suggest including a reference to https://doi.org/10.5194/gmd-12-765-2019, in which JULES + PDM and JULES + TOPMODEL are calibrated for some river catchments in the UK.

---

## Author Comment (AC1)

**Review of 'Evaluating the Atibaia River Hydrology using JULES6.1'**

This paper presents results of applying JULES vn 6.1 to the Atibaia river catchment in Brazil. The authors compare modelled river flow predictions with observed values, comparing the two saturation excess schemes available in the JULES code; PDM and TOPMODEL. The authors perform a sensitivity analysis of parameters within PDM and TOPMODEL and conclude that either option can be used to satisfactorily estimate river flow. The authors suggest that better rainfall drivers would improve the model results further. While this is likely true, I feel that consideration of other sources of error would also be appropriate – in particular the relatively coarse resolution of the JULES runs and the choice of river routing model. I also think the manuscript would benefit from some clarifications, particularly in the results and discussion section. Please see below for specific comments.

**Comments:**

● Please make sure the citation style is consistent throughout.

Reference style checked, updated to APA 7th.

● Line 11 (and elsewhere) the phrase 'environmental processes changes' is a bit confusing – please could you rephrase this?

Rephrased wording to environmental changes

● Line 16. I disagree with the statement that 'Our results show that the coarse resolution of the rainfall data is the main reason to reduce model performance'

Reason for support in the discussion sections:
1. most of flows are well simulated, which means the model regime could simulated the flow although different routing scheme could also alter the model performance.
2. Some intense rainfall event in our stations did not affect the observed flow, which means the rainfall event might not occur in the whole basin.

● Line 34. I'm not clear what the relevance of the MCMC model is here – is this just another example of alternative modelling methods? Please clarify.
Yes. This is just another example of alternative modelling method. I found it not that relevant to the research. I remove this content to make it clear.

● Line 48. I would describe the driving data for JULES as 'meteorological', rather than 'hydrometeorological'.
Description changed.

- Line 56. Does the region really have both C3 and C4 grasses? And, later in the paper you mention that 12% of the catchment is urban. Can you comment on that here?

Yes. The classification is based on MODIS data reclassified by Houldcroft et. al (2009). There are C3 and C4. I add the description of urban.

- Line 61. Please make sure you are clear in the distinction between precipitation and throughfall.
  Description added.

- Line 64. Could you make it a bit clearer that you have compared results from using PDM and TOPMODEL?

Description added. We evaluated the sensitivity of hydrological parameters and calibrated the model to select the most suitable approach for the study region.

- Line 71. What is the spatial scale over which you have calculated the soil physics parameters from textures in the HWSD? IS it representative of the subcatchments you have used?

The spatial scale of HWSD is 0.0833 degree. The sub-basin is classified to two soil categories. We use the soil data from the highest percentage class "Ferric Acrisols". This soil type is one of the representative soils in this catchment according to survey data as listed in study region section (FAO/IIASA/ISRIC/ISSCAS/JRC, 2012; Ottoni et al., 2018; Rossi, 2017).

- Line 74 (or elsewhere) can you make it clear that you have run JULES with each subcatchment as an effective grid box? The more usual approach is to divide an area of interest into approximately square grid boxes. (I don't think there's any problem with doing it by subcatchment, but it needs to be clear!)

We have run JULES with 18 sub-basins. I have made the statement clear.

- Line 76. Can you make clearer how rainfall data are substituted in the case of missing data? And can you give the reader an idea of how representative the rainfall data are likely to be for each catchment?

We use the rainfall data from the nearest station of each gridbox. Monitoring station of Campinas-IAC (Campinas, Atibaia, and Nazare Paulista) has the most comprehensive record, except missing data in 2012 and 2015. We use the rainfall data from DAEE as a substitute in 2012 and 2015. The DAEE data is not the comprehensive (with missing data) in other years, that's the reason we use Campinas-IAC data as the major data source.

- Line 80. Can you justify using this simple river routing model rather than RFM or similar, as is more usual? Should the equation in this paragraph be labelled equation (1)? Also please make sure you define all the symbols in all equations in the paper – adding in units can also be really helpful to the reader. Should there be two different values of C for surface and subsurface flows? How was the value for this chosen?

For C values. Yes. There should be two different values. A lower C should be used for

subsurface flow. However, it has little affect on the results since the change of subsurface flow in JULES is far slower than the surface flow.

● Line 98. Can you make this clearer?

I rewrite this: The study region's rainfall presents a seasonal pattern with rainy summer and dry winter …

● Line 108. Can you explain how you account for the effect of dams?

Description added. L48

The release data of the dams is used as the upstream flow, which is    obtained from the Basic Sanitation Company of the State of Sao Paulo (SABESP, 2020).

● Lin 117. Why do you use a value of 0.085 here? I would suggest that equation (1) is not necessary.

Yes. I didn't find the reason to use 0.085 here. Several values are used for the UK basin, which still need to be investigated. I removed the part of BFI since it is not mention and compared in this research. I change the model evaluation indicator using the NSE, RMSE-observations standard deviation ratio (RSR) , and percent bias ($P_{BIAS}$), following Moriasi et al. (2007).

● Line 120. Which quantity is this the standard deviation of? Please make sure all symbols used are clearly defined – also for the equations from line 125.

Daily flow

● Line 125. Please can you make clear to the reader what these metrics mean – i.e. what constitutes a 'good score' etc?

Content updated. I use indicator from Moriasi et al., 2007 to replace the original content.

● Line 134 (and elsewhere). Can you please make clearer what you mean by 'gradually alters the hydrograph'?

I removed this description due to the rewrite modelling results and discussions. However, I have added Figure 4 to show the change on hydrograph with using different parameters.

● Line 137. You state that the flow is changed to a subsurface flow-dominated regime, but is this shown in the results section?

I have added Figure 4 to show the change on hydrograph.

- Line 139. The choice of b_pdm = 0.5 is also not demonstrated in the results shown.

Description added in L134:

We examined the model performance with a combination of soil depth, shape factor, and minimum soil water content, and found that the highest performance with combination dz=1.0, b=0.5, s=0, which altered the shape factor alone.

- Figure 3. Are these points mean flows? Daily/hourly data points? Please clarify. I would also suggest adding a 1:1 line and/or plotting these with a square aspect ratio to make the relationship clearer. The caption of 3.a is also a little tricky to read.

Mean daily flow. I have made a new square figure 3.

- Line 143. Is this conclusion shown in figure 3d as stated? Please make this clearer if so.
- Line 143. Please justify the statement that peak flow was reduced for a lower ti_max.

ti_max is removed. We added the analysis of fexp with is more relevant. (Martínez-de la Torre et al., 2019)

- Line 150.Please comment on the fact that although TOPMODEL scores better in terms of NSE and R^2, the bias is worse.

I have change the model evaluation indicator using the NSE, RMSE-observations standard deviation ratio (RSR) , and percent bias (P$_{BIAS}$), following Moriasi et al. (2007). In case, TOPMODEL scores better in terms of NSE and RSR. (describe after L141)

- Line 155: Please reword this as the meaning is unclear

I removed this description due to the rewrite modelling results and discussions.

- Line 157 and throughout. Please be specific about which metric you are referring to by 'modelling performance' (e.g. NSE here)

I removed this description due to the rewrite modelling results and discussions.

- Line 160. You state that uncertainty in rainfall data **could** be the main driver for the gap. I agree that it could be, but could you comment on other sources of error and/or uncertainty?

Most of flows are well simulated, which means the model regime could simulated the flow although different routing scheme could also alter the model performance.

● Line 161. What do you mean by 'complimentary' here?

Description added. Replacing the missing data (also added in section 2.1)

● Line 162. Can you rephrase this sentence? Not sure what is meant by 'variated' or 'significant modelling period'.

I have rephrased.

Variated data -> the highly variation of rainfall data

significant modelling period -> most of modelling period

● Line 164 – 167. Can you make sure all of these statements are reflected in the results data shown?

I removed the lines due to the rewrite modelling results and discussions. But I have checked the statements are reflected in the new results data shown.

● Line 169. You compare to SWAT results here; can you show some results or otherwise justify this assertion?

L167. I added the NSE value from the SWAT research.

● **Small things:**

Line 15. Suggest changing 'to evaluate' to 'of evaluating'

Line 49. Suggest changing 'which' to 'and'

Line 54. 'detailly' is not a word, please replace

Line 61. Not sure 'presented' is the right word here – could you reword this?

Line 113. Suggest changing 'modelling' to 'modelled'

Line 130. Suggest replacing 'using' with 'in'.

Relevant changes are made.

● Suggest including a reference to https://doi.org/10.5194/gmd-12-765-2019, in which JULES + PDM and JULES + TOPMODEL are calibrated for some river catchments in the UK.

The relevant paper is now cited for comparison in sensitivity analysis and results section.

---

## Author Comment (AC2)

Comments

● Another highly relevant paper (because it looks at runoff production and riverflow in JULES, and the sensitivity of various runoff parameterisations) is Martínez-de la Torre et al., 2019, Using observed river flow data to improve the hydrological functioning of the JULES land surface model (vn4.3) used for regional coupled modelling in Great Britain (UKC2), Geosci. Model Dev., 12, 765–784, https://doi.org/10.5194/gmd-12-765-2019.

The relevant paper is cited for comparison in sensitivity analysis and results section.

● L31 and following: The phrasing and punctuation make some aspects slightly difficult to follow. Also the final sentence - there are already models, and the perceived need is for one in the public domain (which isn't quite was is said).

● L32: The Sacremento model (SAC-SMA?) will almost certainly have papers that can be cited - assuming this is the same model.

● L34: The discussion of Bayseian and MCMC models is confusing, as neither is used here. Clarify that these are possibilities, not currently used for this catchment.

Due to this comment alongside with the other reviewer's comment, I have removed the less relevant content in this section.

● L74-77: Much of this is about the region, not the model. Move to Sec2.2? Also you talk about sub-catchments before you have introduced us to the whole area (in Sec2.2). I would be tempted to move all the geographical information to before the model is introduced.

I move the geographical information forward. L74-77 is combined to geographical information.

● L74: Somewhere (probably near here) you should clarify that you run the model representing each sub-catchment (Fig.1) as a single model gridbox.

Description added: in gridbox.

● L76: How many data are missing? It would be good to know something about this aspect of data quality. Do you need to swap entire years, or can you just patch data where they are missing (e.g. days or months)?

I replaced the entire year of 2012 and 2015. Over 40% of continuous data are missing (2012/3-2012/8).

- L83-86: This is about your methodology, not JULES. I suggest this might be better in Sec2.3 (which could be renamed).

This is combined to Sec 2.3.

- L86: The TOPMODEL parameterisation in JULES also includes an exponential decline with depth of the saturated hydraulic conductivity, with parameter f (see Gedney and Cox (2003) or Clark and Gedney (2008)). Results are potentially sensitive to that parameter (I expect), so why did you not include that parameter in your sensitivity analysis?

I include parameter f in the sensitivity analysis (JULES code:fexp). I also removed the original content of ti_max since it is less relevant. (Also referring to Martínez-de la Torre et al., 2019)

- L99: Time series - for how many locations? (Fig.1. tells us.)
-

3 stations of Campinas-IAC and 5 stations of DAEE. Also updated in Figure 1.

- L105: Note that higher resolution reanalysis-type datasets are available (e.g. ERA5-based data available via the Copernicus service) - not that that alone guarantees improved accuracy.

We used observed data for air temperature, pressure, humidity. Wind speed and radiation data from reanalysis could affect the model performance. But I believe the change won't be the major part. Also, we haven't got time to parameterize the higher resolution reanalysis-type datasets (ERA5-based data), but will use it in the upcoming research.

- L130: This section only makes much sense if the reader is already familiar with the PDM and TOPMODEL parameteristions in JULES. In general, these parameeristions should be introduced in more detail - e.g. assumptions, how they work, how they differ. The functional forms used should be presented in your manuscript, to save readers having to search through other papers, and so that they can understand how the parameters you vary are used in the model.

Content and functions added in section 2.2 The JULES model.

- The model configuration (parameter values) and the modelling approach should be described in more detail. Suggestions and questions follow: How did you go from the MODIS land cover map to fractions of the model surface types?

Description added: The original 17 land use classes reclassified into 10 JULES land use classes by Houldcroft et al. (2009).

- To what extent is the catchment hydrology modified by human behaviour? If modication is important, is this represented in the model?

The catchment is affected by dam operation. The dam release in the upstream is represented in the model.

- What topographic index data were used for TOPMODEL?

Topographic index data was obtained from (Marthews et al., 2015). The mean value and standard deviation of the basin is used.

- How was the model initialised, and was there any "spin up" period?

The model is spin up using the first year of data (2009). It is allowed to overlap with the main modelling period. Description is added in the content.

- How were all the other parameters and switches set - e.g. did you start from an existing configuration? A keen reader can find all the settings in the Zenodo bundle, but that still doesn't explain where they came from.

There are example data sets which could be installed from the JULES server. Description added in Code and Data availability section.

- Were the optimal parameter settings determined using "expert judgement"? e.g. You present various statistics of the flow, and describe some of the model sensitivity, but how did you come to your final decision? It does not appear to have been through anything such as a weighted-average of the statistics. For some metrics PDM was better than TOPMODEL.

- Was the sensitivity analysis performed "one at a time"? What about any possible interaction between parameters? This possibility should at least be mentioned.

Content Added. We examined the model performance with a combination of soil depth, shape factor, and minimum soil water content, and found that the highest performance with combination dz=1.0, b=0.5, s=0, which altered the shape factor alone.

- You present results only for the flow gauging station that is furthest downstream (I think). Fig.1 suggests that you have two or three other gauging stations that are close to catchment outlets - could you also look at model performance at those points? Those could potentially also tell you if the model behaves better in some parts of the catchment (e.g. headwaters) than elsewhere.

Content Added. The results in compared in the upper, middle, and lower basin.

- Clarify your final parameter settings - i.e. the values that were used for the main runs.

PDM: We found that the highest performance with combination dz=1.0, b=0.5, s=0, which altered the shape factor alone. Therefore, we run the full-time series of modelling with this parameter combination. (L134)

TOPMODEL: In terms of model performance, we found a value of 3.0 simulates the highest NSE (0.61). The value is then selected to be used for the full model simulation. (L138)

General comment on the language - while the manuscript is understandable and written in fairly good English, there are quite a few bits where the language could be improved to make the meaning clearer. If it is possible to get someone (e.g. a native speaker) to spend a bit of time on this, I think you could make improvements without having to spend a lot of effort.

- A few specific examples (just a few phrases that I noted; the more important changes would be about the phrasing of certain sentences):

L50 "JULES's model"

L54 "detailly described"

L70 "physics and chemical properties"

L140: "we found a merely change"

More minor comments

Citations appear in various formats - tidy up.

L202: The reference for Brooks and Corey (1964) looks to have been mangled.

L77: DAEE is (currently) only explained later.

Fig.3: Add units of flow.

Fig.5: Clarify that this is using TOPMODEL.

Relevant modification has been made.

---

## Referee Report (RR1)

**Review of revised manuscript by Chou et al.: Evaluating the Atibaia River Hydrology using JULES6.1**

This manuscript describes the authors' use of the JULES land surface model to simulate the hydrology (principally river flow) of the Atibaia catchment in Brazil. The model is considered to perform reasonably, with the main deficiency being attributed to the lack of good rainfall data for input.

I also reviewed the earlier iteration of this manuscript and I consider the current iteration to be a distinct improvement - the authors have addressed many if not all of the points raised in the previous reviews. The results are now presented and discussed better. Though I do still have some concerns - for example I can imagine further data processing and runs that it would be interesting to carry out - these are largely overridden by my appreciation of what the authors are attempting to do. The main attraction of this work is that it is assessing the extent to which a land surface model can be used to model river flow in a relatively small and relatively data-sparse area that is important for water supply to major urban populations. The need to be able to address pressing issues around water supply to some extent override the desire to provide a definitive modelling study - there is always more that can be done, and sometimes it is better to do a reasonable job rather than demand endless further investigation. That said, I will note a few such possible extensions and questions below.

Abstract L15: "We explore the use of local precipitation collection complement with multiple sources of global reanalysis data". I think this is rather overselling what was done, which was to use local rainfall data with data from a single reanalysis product. Reword as "We explore the use of local precipitation data in conjunction with data from a global meteorological reanalysis" or similar.

Abstract L16: "Our results show that the coarse resolution of rainfall data is the main reason for reduced model performance." As with the previous iteration of the manuscript, I think this is a rather bold conclusion. I would be happier with "Our results suggest...".

**Rainfall data**

L43: "runoff fluxes are simulated in gridbox with rainfall data from the nearest monitoring station (Campinas, Atibaia, and Nazare Paulista) of Campinas Agronomic Institute (Campinas-IAC)". For years with missing data, rainfall "is replaced by using the time series from the nearest Department of Water and Electricity (DAEE) station". As there are 5 DAEE stations shown in Fig.1, I am wondering why you didn't use those for rainfall, rather than using the 3 Campinas-IAC gauges. Perhaps the DAEE stations have more missing data (though they are being used to replace missing data)?

Only gauges from the Atibaia catchment have been used - could gauges from adjacent catchments also be used to get a better idea of the rainfall coverage (perhaps with a more sophisticated analysis than the nearest-gauge approach currently employed)? Given that one of the main conclusions of the study is that better rainfall data are required, it would seem sensible to explore all possible sources of data. Similarly, there are global rainfall products and (meteorological) reanalysis products that might be considered. Although these might be of questionable value over such a relatively small catchment, and their value likely depends on the quality and number of local observations that are incorporated in them, other studies have shown that these can be useful sources of input data - and they can have advantages such as spatial representivity, complete high frequency time series, and consistent relationships between variables. These data might or might not improve the modelling

results, but given the lack of current in situ data, their use should be considered. Ideally this would be part of the current study but otherwise some of these possibilities (or others) should be discussed. The last two paragraphs of Section 3.2 (~L160) could be expanded to better signpost this possible future direction; at present this is scarcely touched with L167 "There is a possibility for the model to be further improved once more adequate rainfall data is available."

**Other points**

L92: "Mean value and standard deviation of the topographic index data is obtained from Marthews et al. (2015) as follows:" - the text that follows actually described something else (related, but not the mean and std dev).

L100: "Soil hydraulic characteristics can be estimated using the relationship of Brooks & Corey (1964) or a more robust formulation of Van Genuchten (1980)." Do the PTFs of Hodnett and Tomasella provide parameter values for both of these hydraulic parameterisations? Which approach was used in the JULES modelling?

Calibration (sensitivity) is assessed only at the basin outlet, but the same parameters are then used for all sub-basins. It would be interesting to know if calibration at other gauges would return similar parameter values (backing up your use of the outlet alone) or might suggest spatial variation of parameters - e.g. from lowland to upland regions, which might be expected to behave differently. Given that several flow gauges are available (and used) why not at least explore whether a better model set up is possible?

**Figures and Tables**

Please indicate the gauging station or part of catchment used in each figure and table. e.g. sensitivity results in Fig.3 is at outlet (I think). L149 says Fig.5 is for the lower basin - this should be included in the caption for Fig.5. I'm guessing that the later plots are also for the outlet - but that should be clear.

Figure 2 doesn't add much - I would consider removing it.

**Minor points and language**

The manuscript is written in reasonable English, but the phrasing is rather odd at times. The meaning is generally obvious, but a fluent speaker of English could tidy the manuscript, possibly with relatively little effort.

Here I list a few examples of bad phrasing here, but there are more:

L25: "Up-to-date, a few research activities"

L28: "In which, a commonly used"

L160: "Despite the highly variation"

L139: "more intense rainfall" - better as "more rainfall". "Intensity" is usually used when characterising shorter timescales, e.g. the rainfall rate during a rain event, not an annual total.

Citations: Some of these are not formatted correctly. e.g. L82 Clark consistently appears as "Clark, Douglas B.".

---

## Author Response (AR2)

- Abstract L15: "We explore the use of local precipitation collection complement with multiple sources of global reanalysis data". I think this is rather overselling what was done, which was to use local rainfall data with data from a single reanalysis product. Reword as "We explore the use of local precipitation data in conjunction with data from a global meteorological reanalysis" or similar.

Statement changed as the suggestions.

- Abstract L16: "Our results show that the coarse resolution of rainfall data is the main reason for reduced model performance." As with the previous iteration of the manuscript, I think this is a rather bold conclusion. I would be happier with "Our results suggest...".

The statement "rainfall data is the main reason for reduced model performance" is removed due to the change on results.

**Rainfall data**

- L43: "runoff fluxes are simulated in gridbox with rainfall data from the nearest monitoring station (Campinas, Atibaia, and Nazare Paulista) of Campinas Agronomic Institute (Campinas-IAC)". For years with missing data, rainfall "is replaced by using the time series from the nearest Department of Water and Electricity (DAEE) station". As there are 5 DAEE stations shown in Fig.1, I am wondering why you didn't use those for rainfall, rather than using the 3 Campinas-IAC gauges. Perhaps the DAEE stations have more missing data (though they are being used to replace missing data)? Only gauges from the Atibaia catchment have been used - could gauges from adjacent catchments also be used to get a better idea of the rainfall coverage (perhaps with a more sophisticated analysis than the nearest-gauge approach currently employed)? Given that one of the main conclusions of the study is that better rainfall data are required, it would seem sensible to explore all possible sources of data. Similarly, there are global rainfall products and (meteorological) reanalysis products that might be considered. Although these might be of questionable value over such a relatively small catchment, and their value likely depends on the quality and number of local observations that are incorporated in them, other studies have shown that these can be useful sources of input data - and they can have advantages such as spatial representivity, complete high frequency time series, and consistent relationships between variables. These data might or might not improve the modelling results, but given the lack of current in situ data, their use should be considered. Ideally this would be part of the current study but

otherwise some of these possibilities (or others) should be discussed. The last two paragraphs of Section 3.2 (~L160) could be expanded to better signpost this possible future direction; at present this is scarcely touched with L167 "There is a possibility for the model to be further improved once more adequate rainfall data is available."

Due to the high variation of rainfall, we thought the data from adjacent may not be an improvement to the model. Instead, we explore the use of the data from 5 DAEE stations in the basin. We use linear regression for the missing data and found that the results could be improved (especially in the year 2017 & 2019). The method is rewrote as "For each sub-basin, the surface ($Q_{surface}$) and sub-surface ($Q_{subsurface}$) runoff fluxes are simulated with rainfall data from the monitoring station (Campinas, Atibaia, and Nazare Paulista) of Campinas Agronomic Institute (Campinas-IAC) and from the station of the Department of Water and Electricity (DAEE, 2022)." The model performance is good at most of the years. Therefore, we thought that rainfall data is not the major source of uncertainty. We removed "rainfall data is the main reason for reduced model performance" and the related statements.

**Other points**
- L92: "Mean value and standard deviation of the topographic index data is obtained from Marthews et al. (2015) as follows:" - the text that follows actually described something else (related, but not the mean and std dev).

Description added "Numerical integration using a two-parameter gamma distribution can be found …"

- L100: "Soil hydraulic characteristics can be estimated using the relationship of Brooks & Corey (1964) or a more robust formulation of Van Genuchten (1980)." Do the PTFs of Hodnett and Tomasella provide parameter values for both of these hydraulic parameterisations? Which approach was used in the JULES modelling?

Soil hydraulic characteristics are estimated using the relationship of Van Genuchten (1980) in our study.

- Calibration (sensitivity) is assessed only at the basin outlet, but the same parameters are then used for all sub-basins. It would be interesting to know if calibration at other gauges would return similar parameter values (backing up your use of the outlet alone) or might suggest spatial variation of parameters - e.g. from lowland to upland regions, which might be expected to behave differently. Given that several flow gauges are available (and used) why not at least explore whether a better model set up is possible?

We explore the sensitivity of hydrological parameters in the upper, middle, and lower basin

in Section 2.3.

"We evaluated the sensitivity of hydrological parameters of PDM and TOPMODEL to determine the most suitable model in the upper (3E-063), middle (3D-006), and lower basin (4D-009) using the simulated results from the first year."

And the results are show in Section 3.1

"We examined the model performance with a combination of soil depth, shape factor, and the minimum soil water content, and found that the highest performance with combination dz=1.0, b=0.5, s=0 in the lower basin, which altered the shape factor alone from the default setup. In the middle and upper basin, an increased value of the minimum soil water content simulated higher performance (dz=1.0, b=0.5, s=0.1). Therefore, we run the full-time series of modelling with these parameter combinations."

**Figures and Tables**
- Please indicate the gauging station or part of catchment used in each figure and table. e.g. sensitivity results in Fig.3 is at outlet (I think). L149 says Fig.5 is for the lower basin - this should be included in the caption for Fig.5. I'm guessing that the later plots are also for the outlet - but that should be clear.

The part of basin is included in the description (Fig 2-7)

- Figure 2 doesn't add much - I would consider removing it.

Figure 2 removed.

- **Minor points and language**

The manuscript is written in reasonable English, but the phrasing is rather odd at times. The meaning is generally obvious, but a fluent speaker of English could tidy the manuscript, possibly with relatively little effort.

Here I list a few examples of bad phrasing here, but there are more:

L25: "Up-to-date, a few research activities"

L28: "In which, a commonly used"

L160: "Despite the highly variation"

L139: "more intense rainfall" - better as "more rainfall". "Intensity" is usually used when characterising shorter timescales, e.g. the rainfall rate during a rain event, not an annual total.

Citations: Some of these are not formatted correctly. e.g. L82 Clark consistently appears as "Clark, Douglas B.".

Relevant changes are made.

**Report #2**

● The authors have nicely addressed most of the comments from my first review. However, I still disagree with the statement from line 17 that 'Our results show that the coarse resolution of rainfall data is the main reason for reduced model performance.' This is certainly suggested by the results but I don't think the results - as shown here - make the argument sufficiently strongly for the quoted statement to be true.

The statement "rainfall data is the main reason for reduced model performance" is removed due to the change on results.

A few additional points:
● line 43: Suggest replacing 'in gridbox' with 'in each sub-basin' for clarity
   line 44: 'The year' should be replaces with 'Years'
   line 88: 'The parameter is initially set...' - please make it clear which parameter is being referred to, and suggest replacing '0.1/0.5' with '0.1 or 0.5' or similar
   line 99: Suggest 'We evaluated the sensitivity of modelled streamflow to the hydrological parameters shown in Table 1..'

Relevant changes are made.
● line 101: Which soil physics scheme did you use here?

Description added: Soil hydraulic characteristics are estimated using the relationship of Van Genuchten (1980).

● line 105: The sentence 'For each basin...' is a bit confusing
Description added: For each sub-basin…

● line 107: The notation in equation (3) needs some clarification - e.g. what is n here? And what does the (t-ti) notation represent? Similarly, make sure all symbols and notation are explained in eqns (4) - (6).

Description added

● Fig 3a. Suggest making sure all the values of dz are in size order in the legend to make

interpretation easier. Also please make clear in the caption that these are daily mean flow values in figure 3.

Order of values adjusted, and "daily flow" are added.

- Line 126: Should '4' be replaced with '0.4'?

Yes. The value is replaced.

- Line 132: '..which altered the shape factor alone.. ' - does this refer to an alteration from the default values? Please clarify.

Description added: which altered the shape factor alone from the default setup.
-
  Line 145: 'Several authors...' but only one reference given. Are there others?

Statement removed due to the updated results, which shows improving performance.

- Fig 6: please label panels a, b and c.

Labelled (New fig 6 and the others)
- Line 154: '..from..'?
  Line 159: Replace 'expect' with 'except'
  Line 165: Please clarify which gap you are referring to here.

These sentences are removed due to the rewrote discussions.

- Line 166: Is the quoted NSE here for the SWAT model or your work? Suggest presenting both values here.

Description added: "The model performance for daily flow in our study (NSE=0.74) is higher than the SWAT model's estim0ation (NSE=0.61 in the validation period) (dos Santos et al., 2020)."

---

## Author Response (AR3)

**Comments revised as marked in the tracked document.**

Comments

This paper is now much clearer - just a few technical suggestions below, which I suggest would further enhance readability.

line 31 suggest '...model structure, and availability..'

line 43: perhaps state number of sub-basin (and state if shown in fig 1) for clarity

line 76: sub-gird -> sub-grid

line 107: suggest: 'The delay time, ti, for each sub-catchment is given by dividing the distance from the sub-catchment to the outlet, di, by....'

line 111: 'is evaluated' -> 'are evaluated'

line 112: suggest: 'Following this sensitivity analysis, the full model is run with...'

line 117: would be slightly clearer if sum is shown from t=0 to N in eqns (4),(5) and (6), and suggest explicitly stating what N is - i.e. over what time period are these calculated? Also, in equation (6) a more usual definition for Pbias is e.g.   . Is there a typo on the top line of (6) or is an alternative formulation used here?

line 126: suggest '..more water held in the soil..'

line 128: sentence starting 'We found that...' a little unclear

line 150: not sure what is meant by the sentence 'In terms of water balance....'

line 165: suggest to clarify '.... since the under/overestimated rainfall in a single site could be amplified'

line 173: spatial and/or temporal variation?

line 174: final sentence a little unclear

---

## Author Response (AR4)

**Author's Response**

The color of figure 1 is change according to the remarks from the preceding review file validation.